# Ocular Paraneoplastic Syndromes

**DOI:** 10.3390/biomedicines8110490

**Published:** 2020-11-10

**Authors:** Joanna Przeździecka-Dołyk, Anna Brzecka, Maria Ejma, Marta Misiuk-Hojło, Luis Fernando Torres Solis, Arturo Solís Herrera, Siva G. Somasundaram, Cecil E. Kirkland, Gjumrakch Aliev

**Affiliations:** 1Department of Ophthalmology, Wroclaw Medical University, Borowska 213, 50-556 Wrocław, Poland; joanna.przezdziecka.dolyk@gmail.com (J.P.-D.); misiuk55@wp.pl (M.M.-H.); 2Department of Optics and Photonics, Wrocław University of Science and Technology, Wyspiańskiego 27, 50-370 Wrocław, Poland; 3Department of Pulmonology and Lung Oncology, Wrocław Medical University, Grabiszyńska 105, 53-439 Wrocław, Poland; anna.brzecka@umed.wroc.pl; 4Department of Neurology, Wroclaw Medical University, Borowska 213, 50-556 Wrocław, Poland; mejma@interia.pl; 5The School of Medicine, Universidad Autónoma de Aguascalientes, Aguascalientes 20392, Mexico; lfts99@yahoo.com.mx; 6Human Photosynthesis© Research Centre, Aguascalientes 20000, Mexico; comagua2000@gmail.com; 7Department of Biological Sciences, Salem University, Salem, WV 26426, USA; siva.somasundaram@salemu.edu (S.G.S.); EKirkland@salemu.edu (C.E.K.); 8Sechenov First Moscow State Medical University (Sechenov University), St. Trubetskaya, 8, bld. 2, 119991 Moscow, Russia; 9Research Institute of Human Morphology, Russian Academy of Medical Science, Street Tsyurupa 3, 117418 Moscow, Russia; 10Institute of Physiologically Active Compounds, Russian Academy of Sciences, Chernogolovka, 142432 Moscow, Russia; 11GALLY International Research Institute, 7733 Louis Pasteur Drive, #330, San Antonio, TX 78229, USA

**Keywords:** cancer-associated retinopathy, melanoma-associated retinopathy, cancer-associated cone dysfunction, paraneoplastic syndromes, vitelliform maculopathy, optic neuritis, uveal melanocytic proliferation, extracellular vesicles

## Abstract

Ocular-involving paraneoplastic syndromes present a wide variety of clinical symptoms. Understanding the background pathophysiological and immunopathological factors can help make a more refined differential diagnosis consistent with the signs and symptoms presented by patients. There are two main pathophysiology arms: (1) autoimmune pathomechanism, which is presented with cancer-associated retinopathy (CAR), melanoma-associated retinopathy (MAR), cancer-associated cone dysfunction (CACD), paraneoplastic vitelliform maculopathy (PVM), and paraneoplastic optic neuritis (PON), and (2) ectopic peptides, which is often caused by tumor-expressed growth factors (T-exGF) and presented with bilateral diffuse uveal melanocytic proliferation (BDUMP). Meticulous systematic analysis of patient symptoms is a critical diagnostic step, complemented by multimodal imaging, which includes fundus photography, optical coherent tomography, fundus autofluorescence, fundus fluorescein angiography, electrophysiological examination, and sometimes fundus indocyjanin green angiography if prescribed by the clinician. Assessment of the presence of circulating antibodies is required for diagnosis. Antiretinal autoantibodies are highly associated with visual paraneoplastic syndromes and may guide diagnosis by classifying clinical manifestations in addition to monitoring treatment.

## 1. Introduction

Sawyer et al. reported the first report of systemic cancer causing visual deterioration and retinal changes in 1976. This opened a new era of research into ocular paraneoplastic syndromes (OPNS). Surprisingly, strict diagnostic criteria remain to be developed. The reason is perhaps the various presentations of OPNS, such as paraneoplastic retinopathy, paraneoplastic optic neuropathy, and paraneoplastic tonic pupils. However, the majority of Paraneoplastic Syndrome (PNS) occur when immune-mediated cross-reactivity involving tumor antigens causes collateral damage to normal host tissues. Alternatively, there are PNS that appear to be caused by the ectopic production of hormones or growth factors that act at a great distance from their production site (Figure 1) [1]. Understanding this main division contributes to a better understanding of OPNS pathophysiology.

The overall incidence of PNS is estimated at about 10% of neoplastic patients. Darnell et al., similar to De Salvo et al., estimated the incidence of OPNS and neurologic paraneoplastic syndromes to be even lower at 0.01% of cancer patients [2,3].

The aim of this paper is to summarize the clinical symptoms and signs associated with different OPNS. Our database search strategy is discussed in the attached file (Appendix A). After removing duplicated studies, we selected 312 published reports for our analysis. We narrowed our review to publications in the past six years that address clinical evaluation and diagnosis.

## 2. Clinical Evaluation

Table 1 and Table 2 provide a summary of clinical presentations that cover the main types of paraneoplastic retinopathies and neuropathies.

Clinicians must take into account the fact that misdiagnosis potential is increased with other ophthalmological changes, such as age-related macular degeneration, pigment epithelium detachment, vitelliform maculopathy, retinitis pigmentosa, stationary night blindness, intraocular inflammation, glaucomatous optic nerve atrophy, or typical optic neuritis.

Table 3 summarizes underlying neoplasm, circulating antibodies or growth factors, and target cells within the visual system.

## 3. Ocular Paraneoplastic Syndromes

### 3.1. Cancer-Associated Retinopathy

Cancer-associated retinopathy (CAR) is most often a bilateral disease with asymmetric presentation of profound visual loss. The progression rate varies from a few days to several months. In common clinical presentation, it affects both cones and rods. This results in symptoms of cone dysfunction, namely, photosensitivity as well as photopsias (type of visual hallucinations seen as light flashes), prolonged glare, decreased best-corrected visual acuity (BCVA), color discrimination or disturbed color vision, and central scotomas. Rod dysfunction may include night blindness, prolonged adaptation to darkness, and peripheral or ring scotomas [4,5,6,7,8,9,10,11,12,13,14,15,16,17,18]. 

Other clinical symptoms include fundus changes. Initially normal fundus can be registered with time periphlebitis and with mild vitritis. Later stages may reveal arteriolar narrowing, retinal pigment epithelial thinning or molting (“salt-and-pepper” appearance), cells in anterior chamber, sheathing of retinal arterioles, and pallor of the optic disc. In optical coherence tomography, loss of outer retinal structures, including ellipsoid and interdigitation zone, can be registered. Fundus autofluorescence (FAF) reveals hyperautofluorescence macular ring area with hypoautofluorescence of the zone outside the macular ring. Fundus fluorescein angiography (FFA) can be normal or may reveal periphlebitis. Central or ring scotomas should be detected in visual field examination. In electroretinography (ERG), rod- and cone-mediated responses are not recordable or significantly decreased, firstly affecting the a-wave and subsequently lining rapidly to a “flat ERG” [8,11,12,13,14,15,16,17,18,19,20].

Figure 2 illustrates mechanisms that enable autoantibodies to access retinal targets.

The recoverin regulates rhodopsin phosphorylation in a calcium-dependent manner. Mechanisms shown in Figure 2 inhibit normal recoverin function, activate calcium-sensitive endonucleases, degrade caspase substrate Procaspase-3 and Poly (ADP) ribose polymerase (PARP), and fragment DNA. These, in turn, induce apoptosis via the mitochondrial pathway involving the caspase enzymes [4,7,11,21,22,23]. Research on classical complement pathway to evaluate the presence of factors C3 and C9 on photoreceptors and retinal pigment epithelium (RPE) surfaces has found that in normal and affected retinas, the C3 and C9 factors can be seen only in the endothelium of the choroidal blood vessels. Possibly, survival of RPE cells during classical complement pathway activation is due to expression of inhibitory proteins on their surface. Clearly, further research is needed [10].

Some studies have shown that tumor-secreted vascular endothelial growth factor (VEGF) as well as placental growth factor (PlGF) act by the retinal VEGF receptor 1 and induce loss of pericytes within the retinal vasculature (breaking the blood–retinal barrier) (shown in Figure 2). CAR-affected retinas show perivascular lymphocytic invasion, normally without or with minimal inflammation [24]. It can be hypothesized that exosome-associated form of CD95L (Fas ligand/FasL), released probably from the two components of the inner blood–retinal barrier (endothelium and retinal pigmented epithelium), may play a crucial role in regulating immune privilege in the eye [25,26]. After initial access due to the vasculitis process, retinal cell damage exacerbates as the RPE damage process spreads via choroidal vasculature (see also Figure 2) [11,27]. Moreover, several works have reported immunomodulatory action by RPE cells. An intriguing study indicated that small RPE-derived exosomes promoted an immunoregulatory phenotype in monocytes [25,28]. Additionally, human RPE cells are known to release from their apical side αB-crystallin in association with exosomes, which is a negative regulator of both innate and cellular immunity [29,30,31]. All of the functions of RPE described above are restricted during CAR, and this in turn may be key to the exacerbation of photoreceptor damage. 

Taking into account the role of RPE in maintaining the eye as an immune-privileged site, the destruction of retinal pigment epithelium can be crucial for exacerbation of antigen-dependent cytotoxic tissue damage during CAR. Additionally, exosomes are responsible for cell surface removal of complementary immune regulators, including CD46, CD55, and CD59 from RPE. The trigger condition is oxidative stress, making RPE cells more vulnerable to complement attack [32]. Cells are able to protect themselves from excessive complement activation by, but not restricted to, cell membrane molecules, such as CD46, CD55, and CD59. CD46 and CD55 regulate the C3 and C5 convertases, and CD59 prevents the formation of C5b-9 complexes [25,32].

Another main autoantigen is alpha-enolase, which is involved in cellular energy production like many other glycolytic enzymes (such as aldolase, glyceraldehyde-3-phosphate dehydrogenase, and pyruvate kinase M2). In this process, the glucose is metabolized to pyruvate in a chain of enzymatic reactions, resulting in production of cellular adenosine triphosphate (ATP). Retina is a very energy-demanding tissue, and in this respect, blocking of the cellular ability to produce ATP by the autoantibodies could be a signal to cell death. The functioning intracellular ATP production pathway is crucial for cell survival [33]. Apart from the function in the glycolytic process, it has been suggested enolase has a role in initiation of a pathological process by modulating the pericellular and intravascular fibrinolytic system. During this process, the enolase is being transferred from the cytoplasm to the cell surface and here serves as a plasminogen receptor that enhances pericellular plasmin production for cell invasion and destruction. Interestingly, the epitopes responsible for plasminogen binding differs significantly from typical pathogenic epitope found in patients with CAR. This suggest that the last mechanism is of low significance in the pathogenesis of CAR [33].

### 3.2. Cancer-Associated Cone Dysfunction (CACD)

Cancer-associated cone dysfunction, a standalone cone dysfunction, is rarely reported. Clinical symptoms originate only from cone dysfunction without any symptoms or electrophysiological signs of rod system failure. Clinical presentation includes mild to moderate BCVA loss, sudden photosensitivity, and total or subtotal loss of color perception. Patients often report improved visual acuity while wearing sunglasses. The ERG shows only cone response suppression [16,34]. According to some researchers, this type of paraneoplastic retinopathy is a subtype of CAR that is caused mainly by antienolase antibodies. Therefore, CACD is presented with overlapping clinical symptoms with CAR as well as other paraneoplastic retinopathies where antienolase antibodies cause retinal tissue damage with or without optic nerve atrophy [1,35,36].

### 3.3. Paraneoplastic Vitelliform Maculopathy

Paraneoplastic vitelliform maculopathy (PVM), also known as acute exudative paraneoplastic polymorphous vitelliform maculopathy (AEPPVM), is a rare bilateral disease. Its clinical presentation may include hemeralopia (decreased visual acuity during the day), blurring, phosphenes, and halos. Symptom duration ranges from a few weeks to several years with asymmetry between eyes [37,38,39,40,41]. The fundus examination usually reveals yellow-orange vitelliform lesions that may be accompanied by serous retinal detachment (“pseudo-hypopion” appearance). In optical coherent tomography (OCT) scans, the vitelliform submacular deposits on the pigment epithelium elevate the neurosensory retina. In the fundus, fluorescein angiography lesion is presented with a blocking effect on the choroid. In the late phase, the contrast uptake by lesion is recorded [37,39]. The ERG is usually within normal limits, but some reports indicate aberrant nonspecific findings. On the other hand, the electrooculography (EOG) can indicate a pathological Arden ratio of 1.1, similar to what is elicited in Best disease [1,42]. 

The pathogenesis is not fully understood. Scarce reports discuss the presence of certain antiretinal and anti-RPE antibodies in peripheral blood samples. It is hypothesized that autoimmune-mediated retinal pigment epitheliopathy occurs, which is based on the presence of antiperoxiredoxin 3 (anti-PRDX3) antibodies. The PRDX3 protein is a mitochondrial peroxidase that is crucial in protection against cellular oxidative damage. Lack of this protection leads to the accumulation of unprocessed cellular debris and (probably) lipofuscin that macroscopically reveals itself as yellowish subretinal deposits [1]. 

Taking into account the clinical presentation and morphological characteristics, PVM and CACD may be categorized as a group of paraneoplastic retinopathies that affects central vision.

### 3.4. Melanoma-Associated Retinopathy

Melanoma-associated retinopathy (MAR) is a rare condition. In contrast to CAR, with MAR, the neoplasm is already known. MAR is characterized by bilateral presentation with the time-shift between the eyes ranging from weeks to months. Clinical presentations include sudden shimmering, flickering, difficulty with the night vision (nyctalopia), photopsia (pulsating, continuous, or intermittent, often with visual hallucinations), decrease of BCVA (less pronounced than in CAR), sensation of color desaturation or conversely increased contrast (hyperphotosensitivity), and bilateral peripheral vision loss [41,43,44,45]. Progression occurs usually over a few weeks to months but, in rare cases, may be sudden. Usually, eye fundus examination is normal in the majority of cases. In the most advanced cases, signs similar to CAR may be observed, namely, disc pallor, attenuated vascular reflexes, zones of retinal pigment epithelial atrophy (“salt-and-paper”), and vitritis accompanied by vasculitis. OCT scans often present macular atrophy with thinning of the inner retina. Fundus fluorescence angiography is usually normal or, in cases of vasculitis, show vascular diffusion. Typical ERG readings are observed, such as disappearance or microvoltage of b-waves while a-waves remain normal in scotopic conditions. In addition, there may be significant dysfunction of bipolar cells with preservation of photoreceptor function. In some studies, a mild reduction of both a-wave and b-wave amplitudes for both scotopic and photopic waveform dependent systems dysfunction. In EOG, the retinal pigment epithelium dysfunction can be evaluated by a reduced Arden ratio [41,43,44,45,46,47].

MAR is defined by three classical characteristics: (1) symptoms of night blindness (nyctalopia) with positive visual phenomena or visual field defects; (2) reduction in b-wave amplitude in ERG; and (3) presence of serum autoantibodies that are reactive with retinal bipolar cells [43].

### 3.5. Bilateral Diffuse Uveal Melanocytic Proliferation (BDUMP)

Bilateral diffuse uveal melanocytic proliferation is a very rare paraneoplastic syndrome characterized by benign proliferation of uveal tract melanocytes. The bilateral visual loss can be profound and rapidly progressive. Patients with BDUMP experience slow, painless, bilateral (in most cases asymmetric), progressive loss of vision over several months. BCVA can also deteriorate due to secondary causes, such as cataract, glaucoma, or iridocyclitis [48,49,50,51,52,53,54,55,56,57,58,59,60]. The proliferation of melanocytes results in subretinal infiltration and an exudative retinal detachment (an outer retinal damage). Fundus examination presents with multiple, bilateral round or oval red (dark) spots or patches at the level of pigment epithelium that can be usually found in the posterior pole. The examiner should perform detailed examination of the peripheral fundus and the peripheral arterial nonperfusion area. Nummular or dermal loss of retinal pigment epithelium in FFA and funduscopy can be found. In the slit-lamp examination, the iris nodules, pigmented keratic precipitates, anterior chamber, and vitreous cells can be observed. In some cases, conjunctival melanocytic proliferation occurs. Hypo- or hyperautofluorescence characteristic “giraffe-like” pattern lesions can be seen during FAF examination. Similar multifocal hyperfluorescence may be revealed during fluorescein angiography in corresponding lesions. Additionally, exudative retinal detachment occurs in the later phases of FFA. Moreover, in OCT, a diffuse thickening of the uveal tract with multiple elevated pigment and nonpigment uveal melanocytic tumors can be recorded. Deep scans in OCT reveal possible atrophy of the choroidal vasculature, which has to be differentiated from age-related macular dystrophy. There are no specific ERG symptoms. The reduction of scotopic and photopic a- and b-wave amplitude can be observed [48,49,50,51,52,53,54,55,56,57,58,61,62].

Pathophysiologically, there are two main causes of this type of paraneoplastic changes: (1) production of melanocytic growth factors by tumor cells with subsequent release into the circulation and (2) antiretinal autoantibodies (suggested anti-α- hepatocyte growth factor (HGF)) in circulation. The impact of the antiretinal autoantibodies appears to be rare compared to that of growth factors. The known and previously isolated IgG antibody is named cultured melanocyte elongation and proliferation factor (CMEP). In addition, the HGF has been suggested [1,57,61,62]. The damage in the RPE layer results in outer blood–retina dysfunction due to RPE malfunctions, and antiretinal autoantigens can subsequently invade the subretinal space [48,57,61].

### 3.6. Paraneoplastic Optic Neuropathy

Paraneoplastic optic neuropathies (PON) are rarer compared to other retinopathies. Classical clinical characteristic of PON include painless, progressive (ranging from days to weeks), bilateral BCVA loss or visual field sensitivity deterioration caused mainly by loss of neural retina (retinal ganglion cells and their axons). Funduscopic evaluation reveals changes in the optic nerve head, such as pallor, hypermia, edema, or local dropout of retinal nerve fiber layer (RNFL) [63,64,65,66,67,68,69,70,71,72,73,74,75].

## 4. Ancillary Tests

The following antibodies should be included in the evaluation of OPNS: antirecoverin, antienolase, anti-GAPDH, and antitransducin. There are several well-known mechanisms of action of these autoantibodies. 

Antirecoverin antibodies act in the mechanism of apoptosis mediated by caspase-dependent pathways along with intracellular calcium influx [11,21,22,23,76]. The main destructive effects are directed to the photoreceptors [1]. 

Antienolase antibodies appear to act via similar mechanism as antirecoverin, but this conjecture needs further investigation [22,36]. The alpha-enolase is engaged in the process of glycolysis in retinal photoreceptors and some other cells within retinal tissue (e.g., Muller cells). The enolase is one of the most common autoantigens. Additionally, it has been reported in several systemic autoimmune, connective tissue and inflammatory disorders, such as Behcet disease, Hashimoto’s encephalopathy, associated antineutrophil cytoplasmic antibody (ANCA)-positive vasculitis, rheumatoid arthritis, systemic lupus erythematosus, multiple sclerosis, primary sclerosing cholangitis, and inflammatory bowel disease [33]. Antienolase antibody appears to target the ganglion cells and inner retinal layers [1]. 

The tubby-like protein 1 (TULIP1) is characteristic of the rods and cones, participating in normal transport during phototransduction proteins. Anti-TULIP1 antibodies interrupt the phototransduction process, and this leads to accumulation of metabolic products [22].

GAPDH is involved in energy metabolism, cell signaling, and synaptic neurotransmission. Anti-GAPDH antibodies appear to contribute to cell damage due to inhibition of membrane fusion, vesicle transportation, or activation of transcription. The GAPDH also plays a role in the processes of apoptosis and oxidative stress, which are other possible mechanisms by which aberrant functions can lead to cell destruction [22,23]. Antitransducin antibody is focused on the outer and inner segments of photoreceptors as well as the cytoplasm of ganglion cells [1,77,78].

Multimodal imaging is critical to diagnosis [79] along with antiretinal antibody testing for diagnosis and treatment. However, the Ocular Immunology Laboratory in USA offers standard procedures for antibody testing and serves clinicians around a world to help them with diagnosis and to monitor treatment (https://www.ohsu.edu/casey-eye-institute/ocular-immunology-lab). The clinician should include fundus photography, OCT with choroid visualization, FFA, FAF, ERG, and multifocal ERG (mf ERG). However, there is no reliable research available on large scale studies for multimodal imaging. There is a great need to conduct a multicenter large data analysis that will include all additional modalities in clinical presentation, such as OCT, FFA, FAF, indocyanine green angiography, and others. 

During the past few decades, a wide range of autoantibodies and molecular markers of OPNS have been proposed and evaluated (Table 3 presents a comprehensive list). Unfortunately, many of them have overlapping characteristic as described above, and the visual system appears to be normal in some cases [80]. The evaluation of molecular markers may aid in the diagnosis of an OPNS. However, this is not an absolute requirement since, certain autoantibodies, growth factors, and other molecules are produced in the absence of PNS.

### 4.1. Antiretinal Antibodies

There are auto-anti-retinal proteins associated with visual symptoms and structural disturbances of retina in paraneoplastic and nonparaneoplastic autoimmune retinopathies [20]. Interestingly, there are some similarities in the clinical presentation of retinopathy associated with specific autoantibodies.

In 2018, Adamus [20] indicated that there was a clinical phenotype for antirecoverin-associated retinopathy that is different from antienolase-associated changes. Differences include onset, ocular symmetry, and clinical presentation, as well as prognosis of the best corrected visual acuity and electrophysiological changes. In some cases, the autoantibodies are strongly associated with specific cancer and precedes its diagnosis. The clinical presentations also vary with respect to the number of cells affected. There is probably a critical number of cells that have to be destroyed before the symptoms are revealed; if the number of cells is below the threshold count, clinical evidence is obscured by the plasticity of retinal neuronal network or by the “filling-in” phenomenon of the whole visual system. Moreover, there are some similarities between hereditary retinitis and autoimmunological background, such as retinitis pigmentosa [22,24,35,77,81,82]. 

Retinal phenotypes based on the seropositivity comprise the following:(A)AntirecoverinAverage age of onset is over 60 years, and patients are predominately female (female/male ratio = 2:1). This entry is highly symmetric, with acute (sudden) unexplained onset. Patient complaints are usually focused on photopsia, nyctalopia, and photophobia. Main presentation is severe central and peripheral vision loss, sometimes described as “ring scotoma”. During the course of the disease, rapid rod and cone loss is seen, often with vision acuity reduced to light perception or to no light perception in some cases. The ffERG is useful since there is a severe decrease in response of rods and cones, in addition to abnormal multifocal ERG [7,20,22,83].(B)AntienolaseAntienolase antibodies are associated with a subacute or even chronic presentation that is often symmetric. There is a variable central and global visual field loss with gradual and variable rate of visual acuity loss. Usually, the best corrected visual acuity is no better than 20/300 after several years of onset. There is characteristic nonequal dysfunction in cone and rod responses in ERG. The cones are more damaged than the rods with mild to severe abnormal mfERG [22,36,63].(C)AntitransducinThis is characterized as sudden but slowly progressive, symmetric, mild, patchy to global visual field loss with visual acuity loss secondary to the visual field symptoms. In electroretinography, primary scotopic defect (rod function) is found, and in mfERG, both decreased amplitudes and delayed timing is seen [22,77,84]. (D)Anti-CAIIThis is usually a subacute, chronic, often symmetric presentation of paraneoplastic retinopathy with mild vitritis and mild to severe concentric constriction of visual field that occurs with color and visual acuity loss. In ERG, slight subnormal rod response can be found, but mfERG is of little use as it varies significantly from case to case [5,22,85]. (E)Anti-Rab6The average age of onset is around 60 years, and patients are predominately female (female/male ratio = 2:1). Onset is sudden and often symmetric. Main patient complaints are photopsia and nyctalopia. Central and peripheral visual field loss can be found. In full-field ERG, the response of rods and cones are decreased, and severely abnormal mfERG can be found (which can progress over weeks to months) [7].(F)Anti-HSP27This is a type of paraneoplastic retinopathy that occurs in patients around 60 years of age with strong predominance of females over males (females/males = 3:1). Its characteristic can be described as subacute, slowly progressive, and mostly symmetric. Patient complaints are mostly focused on photopsia and nyctalopia with constriction of visual field, occurrence of central scotoma, or enlargement of blind spot. In ERG, generalized loss of response of rods and cones can be recorded [7].(G)Anti-GCAP1The anti-GCAP1 retinopathy occurs equally in males and females. Its onset can be characterized as subacute, chronic, and symmetric. The main patient complaint is photophobia. In additional tests, maculopathy can be seen with loss of color vision and visual acuity. Progressive cone loss with gradual cone dysfunction is characteristic in ERG [7].

There are three possible causes of autoretinal antibody formation: (a) immunological response to the tumor; (b) immunological response to the microbiome of infection, including an asymptomatic infection that can trigger antibody formation in susceptible subjects due to conformational mimicry of antigens; and (c) autoimmunological response to the self-antigens that are first sequestrated and separated from the immune system and later released by the damaged retina [24,35,81]. 

The concept of the eye as an immune-privileged site is widely accepted, although its exact mechanisms is not completely understood. It is also known that the phenomenon is not absolute. Retina-specific T cells that are primed for effector function strongly resist conversion to suppressive T regulatory cells (before site-specific local antigen recognition). It is possible that T-effective cells can overcome the immunosuppressive ocular microenvironment to cause primary retinal tissue damage. Considering this scenario, antiretinal antibodies could be secondary causes of retinopathy. More research is needed to distinguish if cellular—rather than humoral—response is responsible for primary retinal tissue damage in paraneoplastic retinopathies [21,24,86,87,88].

### 4.2. Tumor-Expressed Growth Factors

There are known mechanisms of action of different growth factors in paraneoplastic ocular syndrome development. The CMEP factor promotes RPE growth and melanocytic proliferation in the subretinal space. This, in turn, causes disruption of the outer blood–retinal barrier by RPE malfunction [1,48,61]. HGF, probably by prolonged high-level stimulation of this growth factor, combined with retinal autoantibodies may drive choroidal nevis growth and RPE damage [57]. VEGF and PlGF play roles in the reduction of endothelial cells in the retinal macrophages and lymphocytes to invade the retina. The described mechanism depends on damage to or destruction of tight junctions between endothelial cells that cause dysfunction of the inner blood–retina barrier [24].

## 5. Possible Role of Extracellular Vesicles (EVs) in the Development of Paraneoplastic Syndromes

Recently, the concept of the association between paraneoplastic syndromes and extracellular vesicles has emerged. The term EV encompasses exosomes, microvesicles, and apoptotic bodies [89]. EVs enable communication between cancer cells and other organs in the body [90]. For at least the past decade, exosomes have been considered as a factor that plays a potential role in the development of paraneoplastic syndromes [91], mostly in cancers other than those developing in the eye [91,92,93]. Usually, PNS occur when immune-mediated cross-reactivity activates tumor antigens, resulting in damage to normal host tissues. There are PNS that seem to be caused by the ectopic production of a hormone or a growth factor that acts at a great distance from the production site. The exact pathophysiologic mechanisms that contribute to breaking the eye immune privilege, enabling the occurrence of OPNS, have not been precisely described. Although not conclusive, it can be hypothesized that EVs participate in the onset and progression of OPNS.

## Figures and Tables

**Figure 1 biomedicines-08-00490-f001:**
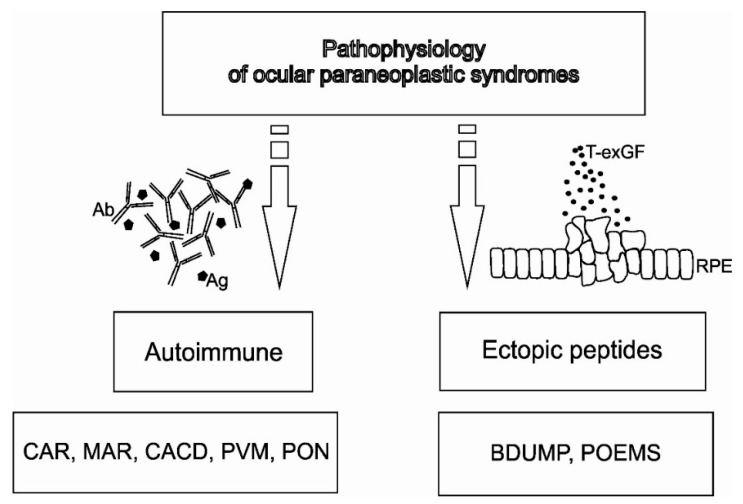
The pathophysiology of ocular paraneoplastic syndromes (OPNS). Autoimmune pathomechanism is presented with cancer-associated retinopathy (CAR), melanoma-associated retinopathy (MAR), cancer-associated cone dysfunction (CACD), paraneoplastic vitelliform maculopathy PVM), and paraneoplastic optic neuritis (PON). Ectopic peptides, caused by tumor-expressed growth factors (T-exGF), are presented with bilateral diffuse uveal melanocytic proliferation (BDUMP) and polyneuropathy, organomegaly, endocrinopathy, monoclonal gammopathy, and skin changes syndrome (POEMS).

**Figure 2 biomedicines-08-00490-f002:**
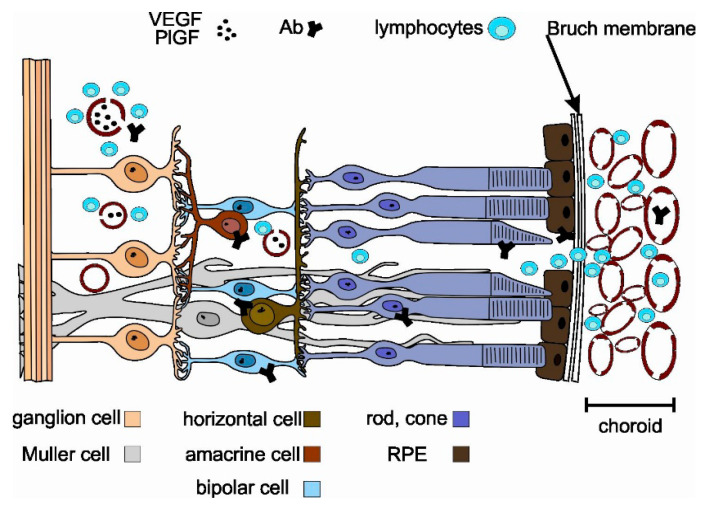
Mechanism of retinal invasion of autoantibodies and lymphocytes during CAR.

**Table 1 biomedicines-08-00490-t001:** Summary of clinical features in different types of paraneoplastic retinopathies based on the included articles.

Clinical Features	CAR	CACD	PVM	MAR	BDUMP
Onset	Acute, sudden (few days to several months)	Subacute	Acute or subacute (few weeks to several years)	Acute (few weeks to months), may be sudden	Acute, sudden (several months)
Ocular symmetry	Bilateral with asymmetric presentation	Often symmetric	Bilateral with asymmetric presentation	Bilateral	Bilateral with asymmetric presentation
Photosensitivity	+++	+++	−	−	−
Photopsias	+++	+	−	+++	−
Glare	+++ prolonged	−	++	−	−
Halo	−	−	+++	−	−
Starburst	−	−	−	−	−
Color discrimination problems (basic colors	++	+++	−	−	−
Disturbed color vision (color desaturation)	++	+++	−	++	−
Night blindness	+++	−	−	+++	−
Prolonged adaptation to darkness	+++	−	−	+	−
Improvement of visual acuity while wearing sunglasses	−	+++	−	−	−
Significant decrease of visual acuity during the day	−	+	+++	−	−
Phosphenes (visual hallucinations)	+++	−	+++	++	−
Sudden shimmering	−	−	+	+++	−
Sudden flickering	++	−	−	+++	−
Increased contrast sensitivity (hyperphotosensitivity)	−	−	−	+++	−
Increased color contrast sensitivity (hyperphotosensitivity)	−	−	−	+	−
Pain of the eye	−	−	−	−	+/−
Feeling of “full” eyes	−	−	−	−	+/−

CAR—cancer-associated retinopathy; CACD—cancer-associated cone dysfunction; PVM—paraneoplastic vitelliform maculopathy; MAR—melanoma-associated retinopathy; BDUMP—bilateral diffuse uveal melanocytic proliferation; “−”—absent, “+/−”—possibly present, “+”—present, “++”—strongly associated, “+++”—characteristic to this entry.

**Table 2 biomedicines-08-00490-t002:** Summary of clinical work-up results in different paraneoplastic retinopathies based on the included articles.

Clinical Work-Up	CAR	CACD	PVM	MAR	BDUMP
BCVA	Severely decreased	Mildly to moderately decreased	Blurred vision, mildly decreased	Mildly to severely decreased	Moderately to severely decreased
Visual field	Central or ring scotoma, peripheral scotomas	Central scotoma	Central/paracentral scotoma	Central/paracentral scotoma	Nonspecific
OCT signs	Loss of outer retinal structures, including ellipsoid and interdigitation zone (central and peripheral)	Central loss of outer retinal structures, including ellipsoid and interdigitation zone with normal periphery	Vitelliform submacular deposits on the pigment epithelium that elevate the neurosensory retina	Macular atrophy with thinning of the inner retina	Diffuse thickening of the uveal tract with multiple elevated pigment and nonpigment uveal melanocytic tumors, possible atrophy of choroidal vasculature in Enhanced Depth Imaging (EDI) scans was observed
FAF	Ring macular hyperautofluorescence with surrounding hypoautofluorescence	Nonspecific	Nonspecific	Nonspecific	Hypo-/hyperautofluorescence characteristic “giraffe-like” pattern lesions
FFA	Normal/periphlebitis	Nonspecific	Blocking effect on the choroid with late-phase contrast uptake	Normal/vasculitis with vascular diffusion	Peripheral arterial nonperfusion area, nummular or dermal loss of retinal pigment epithelium, exudative retinal detachment in the late phases
ICGA	Normal	Normal	Normal	Normal	Normal/atrophy of choroidal vasculature
Full-field ERG	Rods and cones equally affected, firstly affecting the a-wave, progression to the “flat ERG”	Affected only cone response	Nonspecific, variable results	Scotopic—disappearance or microvoltage of b-wave, normal a-wave. Dysfunction of bipolar cells	Reduction of scotopic and photopic a- and b-wave amplitude
mfERG	Severely abnormal	Partially abnormal	Nonspecific	Mildly abnormal	Mildly abnormal
EOG	Normal Arden ratio	Normal Arden ratio	Variable Arden ratio	Reduced Arden ratio	Normal Arden ratio

CAR—cancer-associated retinopathy; CACD—cancer-associated cone dysfunction; PVM—paraneoplastic vitelliform maculopathy; MAR—melanoma-associated retinopathy; BDUMP—bilateral diffuse uveal melanocytic proliferation; BCVA—best corrected visual acuity; OCT—optical coherence tomography; FAF—fundus autofluorescence; FFA—fundus fluorescein angiography; ICGA—indocyjanin green angiography; ERG—electoretinography; ffERG—full field electroretinography; mfERG—multifocal electroretinography; EOG—electrooculography.

**Table 3 biomedicines-08-00490-t003:** Summary of underlying neoplasm, circulating antibodies or growth factors, and target cells within the visual system.

OPNS	Neoplasm	Mediator *	Target Cell
CAR	Small-cell lung carcinoma, other lung neoplasm, breast cancer, cancers of the cervix, ovary, uterus and thymus, osteosarcoma, Warthin tumor of parotid gland, prostate, pancreatic neuroendocrine, small bowel, bladder and laryngeal neoplasms, lymphomas (systemic follicular cell lymphoma), and colon adenomas	Recoverin, retinal enolase, TULP1, hsc-70 and 60, AIPL1, IRBP, PNR, GAPDH, aldolase C, transducin-α, GCAPs, HSP27 and Rab6A, CA II, CRMP5, antiretinal autoantibodies against arrestin and 64-kDa and 94-kDa, C3, and C9	Rod, cone, bipolar cell, retinal pigment epithelium
CACD	small-cell endometrial cancer, primary cervical intraepithelial neoplasia, occult small cell lung carcinoma, and laryngeal carcinoma	Recoverin, retinal enolase, and protein whose molecular weight is 50 and 40 kDa	L and M than S cones
PVM	cutaneous and mucosal melanoma, lymphoma	PRDX3, ROS, bestropin-1, CA II, IRBP, proteins 35-kDa and 68-kDa	Probably cones, bipolar cells, and rods
MAR	cutaneous and mucosal melanoma	TRPM1, MLSN1 α-enolase, recoverin or hsc-70, CA II, IRBP, bestrophin, myelin basic protein, mitofilin, titin, and rod outer segment proteins	Bipolar cells (preferably ON-bipolar cell)
BDUMP	ovarian, cervix, uterus, colon and rectum cancer, gallbladder cancer, neoplasm of the retroperitoneal space, and a variety of lung cancers	CMEP factor; AAbs against 35-kDa, 46-kDa, 30-kDa, 50-kDa, and 70-kDa proteins; α-HGF and HGF	Pigment epithelium
PON	adenocarcinoma and small-cell carcinoma of the lung, prostate carcinoma, stomach carcinoid tumor, colon adenocarcinoma, cutaneous melanoma, occult pancreatic nonsecretory neuroendocrine tumor, thymoma	CRMP5, aquaporin 4, MBP, ANNA-1, recoverin, enolase	Photoreceptors, ganglion cells, and their axons

* Substance (antibody, growth factor, or peptide) that mediates the reaction in the target cell/cells; CAR—cancer-associated retinopathy; CACD—cancer-associated cone dysfunction; PVM—paraneoplastic vitelliform maculopathy; MAR—melanoma-associated retinopathy; BDUMP—bilateral diffuse uveal melanocytic proliferation; PON—paraneoplastic optic neuritis; TULP1—tubby-like protein 1; hsc-70—heat shock cognate protein 70; hsc-60—heat shock cognate protein 60; AIPL1—aryl hydrocarbon receptor interacting protein-like 1; IRBP—interphotoreceptor retinoid binding protein; PNR—photoreceptor cell-specific nuclear receptor; GAPDH—glyceraldehyde 3-phosphate dehydrogenase; GCAPs—guanylyl cyclase-activating proteins; HSP27—heat shock protein 27; Rab6A—Rab6A GTPase; CRMP5—collapsin response mediator protein 5; CA II—carbonic anhydrase II; PRDX3—peroxiredoxin 3 (26-kDa); ROS—rod outer segment protein (120-kDa); TRPM1—transient receptor potential cation channel, subfamily M member 1 (that is labeled on ON-bipolar cells); MLSN1—melastin 1; CMEP factor—cultured melanocyte elongation and proliferation factor; HGF—hepatocytes growth factor; ANNA-1—type 1 antineuronal nuclear antibody; MBP—myelin binding protein.

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
