# Peer review of "Ocular Paraneoplastic Syndromes"

_biomedicines, 2020, doi:10.3390/biomedicines8110490_

Round 1

Reviewer 1 Report

Przezdziecka—Dolyk et al (+ 9 coauthors) overviewed the literature on visual paraneoplastic syndromes related to retina and optic nerve. This manuscript provides some useful information on recent cases but tends to reference a lot of published reviews. At the same time, they missed out recently published papers regarding the mechanism of autoimmune retinopathies.  There are more concerns that have to resolve.

  1. This review is no well-balanced – CAR is the syndrome that we know the most about clinical presentation, findings, and autoimmune pathology. However, they focused on uncommon syndromes with a lesser number of publications.
  2. They claimed that “ectopic peptides” are involved in pathology of paraneoplastic retinopathies. However, there is a little evidence that they play a role in CAR or MAR and others.
  3. Table 1 and 2 – how the data was collected? Explain the significance of pluses: +, ++, +++.
  4. Table 3 – contains a list “circulating mediators” but in fact, it is a list of autoantigens. There are autoantibodies that are found in circulation NOT those antigens. Some of them are not related to CAR, e.g. Anti-Hu.
  5. Many references are wrong – e.g. page 6 line 125 and 133, too many to point out. They should carefully evaluate the appropriateness their citations.
  6. What is the purpose of a clinical questionnaire or clinical evaluation chart in this review?
  7. Table 4 has been copied for a published paper, which NOT acceptable.
  8. Page 14 line 358 – chapter 5 does not seem be related to the previous parts of this review.
  9. The title is inappropriate to the content of this manuscript.

Author Response

 Dear Reviewer 1,

On behalf of my coauthors, please accept my sincere thanks and gratitude for careful perusal and critical review of our manuscript entitled:  OCULAR PARANEOPLASTIC SYNDROMES (PREVIOUS TITLE: Ocular paraneoplastic syndromes: clinical pearls and pitfalls). We have revised the manuscript based upon the reviewers’ comments as well as your suggestion. Adequate care has been taken to accommodate each and every suggestion of the reviewers. An itemized, “point-by-point” reply to all the comments is attached separately where we have clearly presented our specific response and additions, deletions and/or modifications that have been made in the revised text, and highlighted.

  1. ….. missed out recently published papers regarding the mechanism of autoimmune retinopathies

Response: Thank you for the remark. Missing published papers have been added into the body of revised manuscript.

  1. This review is no well-balanced – CAR is the syndrome that we know the most about clinical presentation, findings, and autoimmune pathology. However, they focused on uncommon syndromes with a lesser number of publications.

Response: According to the Reviewer’s suggestion the following paragraphs have been added in order to maintain better balance of the paper:

The alpha-enolase in engaged in the process of glycolysis in retinal photoreceptors and some other cells within retinal tissue (e.g. Muller cells). The enolase is one of the most common autoantigens, additionally it has been reported in several systemic autoimmune, connective tissue and inflammatory disorders such as: Behcet disease, Hashimoto’s encephalopathy, ANCA-positive vasculitis, rheumatoid arthritis, systemic lupus erythematosus, multiple sclerosis, primary sclerosing cholangitis and inflammatory bowel disease [33].

The tubby-like protein 1 (TULIP1) is characteristic for the rods and cones, participates in normal transport during phototransduction proteins. Anti –TULIP1 antibodies interrupt phototransduction process and this leads to accumulation of metabolic products [22].

  1. They claimed that “ectopic peptides” are involved in pathology of paraneoplastic retinopathies. However, there is a little evidence that they play a role in CAR or MAR and others.

Response: In respect to the ectopic peptides (in the introduction we just state the main division of the pathophysiology of ocular paraneoplastic retinopathies) we indicate the research that is associated with some ocular paraneoplastic retinopathies in the fragment describing pathogenesis of BDUMP).

  1. Table 1 and 2 – how the data was collected? Explain the significance of pluses: +, ++, +++.

Response: In the title of the tables “based on the included articles”  has been added. 

  1. Table 1 and 2 –Explain the significance of pluses: +, ++, +++.

Response: According to the Reviewer’s  indication the explanation of the  significance of pluses have been added:

“-” - absent, “+/-” possibly present, “+” - present, “++”-strongly associated, “+++” - characteristic to this entitle

  1. Table 3 – contains a list “circulating mediators” but in fact, it is a list of autoantigens. There are autoantibodies that are found in circulation NOT those antigens. Some of them are not related to CAR, e.g. Anti-Hu.

Response: Table has been  corrected, accordingly. Now:

Table 3 summarizing underlying neoplasm, circulating antibodies or growth factors and target cells within the visual system)

  1. Many references are wrong – e.g. page 6 line 125 and 133, too many to point out. They should carefully evaluate the appropriateness their citations.

Response: Thank you for finding these mistakes. The citations have been corrected.

  1. What is the purpose of a clinical questionnaire or clinical evaluation chart in this review?

Response: According to this remark both questionnaire and chart have been removed.

  1. Table 4 has been copied for a published paper, which NOT acceptable.

Response: According to this remark the whole table has been removed. Instead, the paragraph  has been added:

Differences include onset, ocular symmetry, clinical presentation, as well as prognosis of the best corrected visual acuity and electrophysiological changes. In some cases, the autoantibodies are strongly associated with specific cancer and precedes its diagnosis. The clinical presentations also vary with respect of number of cells affected. There is probably a critical number of cells that have to be destroyed before the symptoms are revealed: at some number of cells below the threshold count, the clinical evidence is obscured by the plasticity of retinal neuronal network or by the “filling-in” phenomenon of the whole visual system. Moreover, there are some similarities with hereditary retinitis with autoimmunological background such as retinitis pigmentosa[22,24,35,77,80,81].

Retinal phenotypes based on the seropositivity:

  1. Anti-recoverin

Average age of onset is in over 60 years old patients predominately females (female:male ratio = 2:1). This entitle is highly symmetric with acute (sudden) unexplained onset. Patients’ complains usually are focused on photopsia, nyctalopia and photophobia. Main presentation is severe central and peripheral vision loss sometimes described as “ring scotoma”. During the course of the disease the rapid rod and cone loss is seen often with vision acuity reduced to light perception or in some cases to no light perception. The full field ERG is useful as there is equal rods and cones severe decrease in response alongside with severely abnormal multifocal ERG (mfERG)[7,20,22,82].

  1. Anti-enolase

Anti-enolase antibodies are associated with a subacute or even chronic presentation that is often symmetric. There is a variable central and global visual field loss with gradual visual acuity loss with variable rate of the loss. Usually the best corrected visual acuity is no better than 20/300 after several years of onset. There is characteristic non-equal dysfunction in cones and rods responses in ERG. The cone are more damaged than the rods with mildly to severely abnormal mfERG[22,36,63].

  1. Anti-transducin

It is characterized as sudden but slowly progressive, symmetric, mild, pathy to global visual field loss with visual acuity loss secondary to the visual field symptoms. In electroretinography primary scotopic defect (rod function) is found and in mfERG both decreased amplitudes and delayed timing is seen [22,77,83].

  1. Anti-CAII

This is usually subacute, chronic often symmetric presentation of paraneoplastic retinopathy with mild vitritis, mild to severe concentric constriction of visual field that occurred with colour and visual acuity loss. In ERG slight subnormal rods response can be found but mfERG is of little use as it vary significantly from case to case[5,22,84].

  1. Anti-Rab6

In described entitle the average age of onset is in around 60years old patients predominately females (female:male ratio = 2:1) with sudden often symmetric onset. Main patients’ complains are photopsia and bnyctalopia. Central and peripheral visual field loss can be found. In the full field ERG the rods and cones decrease in response alongside with severely abnormal multifocal ERG (mfERG) can be found (can progress over weeks to months) [7].

  1. Anti-HSP27

It is a type of paraneoplastic retinopathy that occurs in patients around 60 years old with strong predominance of females over males (females:males=3:1). It’s characteristic can be described as subacute, slowly progressive and mostly symmetric. Patients’ complains are focused mostly on the photopsia and nyctalopia with constriction of visual field, occurrence of central scotoma or enlargement of blind spot. In ERG generalized rods and cones loss of response can be recorded [7].

  1. Anti-GCAP1

The anti-GCAP1 retinopathy occurs equally in males and females. Its onset can be characterized as subacute, chronic and symmetric. The main patients complain is photophobia. In the additional test the maculopathy can be seen with loss of color vision and visual acuity. Characteristic is a progressive cone loss with gradual cone dysfunction in ERG [7].

  1. Page 14 line 358 – chapter 5 does not seem be related to the previous parts of this review.

Response: According to the suggestion, all the sentences not related to the previous part of the review have been deleted and only the part explaining the hypothesized role of extracellular vesicles in paraneoplastic syndromes remained.

  1. The title is inappropriate to the content of this manuscript

Response: Accordingly to the Reviewers suggestion the title has been shortened to “Ocular paraneoplastic syndromes”

Reviewer 2 Report

This is a well-written, clear review of the ocular paraneoplastic syndromes (OPS). The authors synthesized very well the key points for each of the OPS. However, while dwelling extensively with the mechanism of each OPS, the authors fail to show the importance of both multimodal imaging and functional testing to the diagnosis. A multimodal imaging figure would be interesting to add for each of the discussed OPS, accompanied, if available, by functional testing.

Author Response

DEAR REVIEWER 2,

 On behalf of my coauthors, please accept my sincere thanks and gratitude for careful perusal and critical review of our manuscript entitled:  OCULAR PARANEOPLASTIC SYNDROMES (PREVIOUS TITLE: Ocular paraneoplastic syndromes: clinical pearls and pitfalls). We have revised the manuscript based upon the reviewers’ comments as well as your suggestion. Adequate care has been taken to accommodate each and every suggestion of the reviewers. An itemized, “point-by-point” reply to all the comments is attached separately where we have clearly presented our specific response and additions, deletions and/or modifications that have been made in the revised text, and highlighted.

  1. the authors fail to show the importance of both multimodal imaging and functional testing to the diagnosis.

Response: According to this suggestion in the chapter 4 (Ancillary tests) a paragraph has been enlarged as follows:

Multimodal imaging is critical to diagnosis. The clinician should include fundus photography, OCT with choroid visualisation, FFA, FAF, ERG and multifocal ERG.  However, there is no big data on multimodal imaging original research that we could fully rely on. There is a great need to conduct a multicenter big data analysis that will include in clinical presentation all additional modalities such as OCT, FFA, FAF,  indocyanine green angiography and others.

  1. A multimodal imaging figure would be interesting to add for each of the discussed OPS, accompanied, if available, by functional testing.

Response: We much appreciate this valuable suggestion, but unfortunately, we are not able to include original multimodal imaging figures.

Multimodal imaging is critical to diagnosis. The clinician should include fundus photography, OCT with choroid visualisation, FFA, FAF, ERG and multifocal ERG.  However, there is no big data on multimodal imaging original research that we could fully rely on. There is a great need to conduct a multicenter big data analysis that will include in clinical presentation all additional modalities such as OCT, FFA, FAF,  indocyanine green angiography and others.

On behalf of my coauthors, I once again express our sincere thanks to the esteemed Editors, and erudite reviewers for their valuable suggestions and constructive input to improve the quality of this manuscript.

Round 2

Reviewer 1 Report

The author mostly responded to the reviewers’ concerns. However, the manuscript still require some work. The author should carefully review their manuscript for errors.

  1. Abstract: “Assessment of the presence of circulating antibodies may be helpful but is not necessary and may delay diagnosis” – most experts in CAR and MAR that the syndromes are mediated by antibodies. Therefore testing for antibody is a requirement.
  2. the authors in different parts of the manuscript use the terms “full field ERG”, “multifocal ERG”, “ffERG”, or mfERG”. They should explain the abbreviation first time they use it.
  3. Line 70: “After de-duplication, published reports remained” –explain this sentence.
  4. Fig. 2: I suggest that you explained the mechanism. Just the title is not sufficient and not clear from the cartoon.
  5. Line 156: The lack of functioning intracellular ATP production pathway is crucial for cell survival [33] – this is not correct.
  6. Line 257 “anti-ricoverin,..” – should be anti-recoverin.
  7. Line 353 – the paragraph needs references.

Author Response

Dear Reviewer 1,

On behalf of my coauthors, please accept my sincere thanks and gratitude for careful perusal and critical review of our manuscript entitled:  OCULAR PARANEOPLASTIC SYNDROMES. We have revised the manuscript based upon the reviewers’ comments as well as your suggestion. Adequate care has been taken to accommodate each and every suggestion of the reviewers. An itemized, “point-by-point” reply to all the comments is attached separately where we have clearly presented our specific response and additions, deletions and/or modifications that have been made in the revised text, and highlighted.

Point-by-Point Answers to the Comments

Reviewer 1:

Comments and Suggestions for Authors

The author mostly responded to the reviewers’ concerns. However, the manuscripts still require some work. The author should carefully review their manuscript for errors.

Response: The Authors thank the reviewer for extended comments.

Abstract: “Assessment of the presence of circulating antibodies may be helpful but is not necessary and may delay diagnosis” – most experts in CAR and MAR that the syndromes are mediated by antibodies. Therefore testing for antibody is a requirement.

Response:: We executed the corrections as per the suggestions.

the authors in different parts of the manuscript use the terms “full field ERG”, “multifocal ERG”, “ffERG”, or mfERG”. They should explain the abbreviation first time they use it.

Response: We executed the corrections as per the suggestions not repeating the full abbreviations.

Line 70: “After de-duplication, published reports remained” –explain this sentence.

Response:  We executed the corrections as per the suggestions. New sentence “Assessment of the presence of circulating antibodies is necessary since CAR and MAR syndromes are mediated by antibodies”.

Fig. 2: I suggest that you explained the mechanism. Just the title is not sufficient and not clear from the cartoon.

Response:  We executed the corrections as per the suggestions. New sentence “This cartoon explains the autoantibody and lymphocytes migrations through ganglion, bipolar cells, rod, cone and reaches choroid tissues through brush membrane.”

Line 156: The lack of functioning intracellular ATP production pathway is crucial for cell survival [33] – this is not correct.

Response: We executed the corrections as per the suggestions. New sentence” The functioning intracellular ATP production pathway is crucial for cell survival [33]”.

Line 257 “anti-ricoverin,..” – should be anti-recoverin.

Response:  We executed the corrections as per the suggestions.

Line 353 – the paragraph needs references.

Response: We executed the corrections as per the suggestions’ new reference. ((https://www.ncbi.nlm.nih.gov/pmc/articles/PMC5911469/

Reviewer 2 Report

I would like to thank the authors for making (some of) the requested changes. 

Author Response

Dear Reviewer 2,

 On behalf of my coauthors, please accept my sincere thanks and gratitude for careful perusal and critical review of our manuscript entitled:  OCULAR PARANEOPLASTIC SYNDROMES. We have revised the manuscript based upon the reviewers’ comments as well as your suggestion. Adequate care has been taken to accommodate each and every suggestion of the reviewers. An itemized, “point-by-point” reply to all the comments is attached separately where we have clearly presented our specific response and additions, deletions and/or modifications that have been made in the revised text, and highlighted.

Point-by-Point Answers to the Comments

Reviewer 2 Comments:

Comments and Suggestions for Authors

I would like to thank the authors for making (some of) the requested changes.

Response: We thank the reviewer for the encouraging comments and we executed the corrections as per the suggestions.

Round 3

Reviewer 1 Report

Abstract line 41 “Assessment of the presence of circulating antibodies is required but is not necessary for the diagnosis.” – This contradictory statement. Anti-retinal autoantibodies are highly associated with visual paraneoplastic syndromes. Testing for autoantibodies against anti-retinal autoantibodies may guide diagnosis, classify clinical manifestations in disease entities and is useful to monitor treatment.

Line 289: “The lack of an accepted standard procedure of anti-retinal antibodies is still a problem” – is no clear why the authors claim that there not standard procedures. The authors are not well-informed. The Ocular Immunology Laboratory in USA offers standard procedures for antibody testing and serves clinicians around a word to help them with diagnosis and monitoring the treatment.

Author Response

Dear Reviewer 1:

On behalf of my coauthors, please accept my sincere thanks and gratitude for careful perusal and critical review of our manuscript entitled:  OCULAR PARANEOPLASTIC SYNDROMES. We have revised the manuscript based upon the reviewers’ comments as well as your suggestion. Adequate care has been taken to accommodate each and every suggestion of the reviewers. An itemized, “point-by-point” reply to all the comments is attached separately where we have clearly presented our specific response and additions, deletions and/or modifications that have been made in the revised text, and highlighted.

Point-by-Point Answers to the Comments.

COMMENTS:

Comments and Suggestions for Authors

Abstract line 41 “Assessment of the presence of circulating antibodies is required but is not necessary for the diagnosis.” – This contradictory statement. Anti-retinal autoantibodies are highly associated with visual paraneoplastic syndromes. Testing for autoantibodies against anti-retinal autoantibodies may guide diagnosis, classify clinical manifestations in disease entities and is useful to monitor treatment.

Response:  Assessment of the presence of circulating antibodies is required but is not necessary for the diagnosis. Anti-retinal autoantibodies are highly associated with visual paraneoplastic syndromes and may guide diagnosis, classify clinical manifestations in addition to monitor treatment.

Comments:

Line 289: “The lack of an accepted standard procedure of anti-retinal antibodies is still a problem” – is no clear why the authors claim that there not standard procedures. The authors are not well-informed. The Ocular Immunology Laboratory in USA offers standard procedures for antibody testing and serves clinicians around a word to help them with diagnosis and monitoring the treatment.

Response:  Multimodal imaging is critical to diagnosis [79] along with anti-retinal antibodies testing for diagnosis and. treatment The lack of an accepted standard procedure of antiretinal antibodies is still a problem. The seropositivity to the certain auto-antibodies can be helpful but due to inconsistency between different laboratories and lack of an accepted gold standard makes the testing for autoantibodies second best after the clinical presentation with intense multimodal imaging. Lack of However, the Ocular Immunology Laboratory in USA offers standard procedures for antibody testing and serves clinicians around a word to help them with diagnosis and monitoring the treatment.( https://www.ohsu.edu/casey-eye-institute/ocular-immunology-lab) anti-retinal antibodies did not define that patient do not suffer from ocular paraneoplastic syndrome and seropositivity is not enough for proper diagnosis [79].

On behalf of my coauthors, I once again express our sincere thanks to the esteemed Editors, and erudite reviewers for their valuable suggestions and constructive input to improve the quality of this manuscript.

I look forward your favorable decision regarding our manuscript in the near future.

Please do not hesitate to contact me if you have any additional questions.

Yours truly,

Gjumrakch

Dr. Gjumrakch Aliev, MD&PhD

Professor of Cardiovascular, Neuropathology

and Gerontology & President

“GALLY” International Research Institute

7733 Louis Pasteur Drive, #330

San Antonio, TX 78229 USA

Phone: 210-442-8625,  440-263-7461

Email:   [email protected], [email protected]